# Impact of COVID-19 Social Distancing Policies on Traffic Congestion, Mobility, and NO$_2$ Pollution

Alyse K. Winchester [1], Ryan A. Peterson [1,*], Ellison Carter [2] and Mary D. Sammel [1]

1    Department of Biostatistics and Informatics, Colorado School of Public Health, University of Colorado-Denver Anschutz Medical Campus, Aurora, CO 80045, USA; alyse.winchester@cuanschutz.edu (A.K.W.); mary.sammel@cuanschutz.edu (M.D.S.)
2    Department of Civil and Environmental Engineering, Colorado State University, Fort Collins, CO 80523, USA; Ellison.Carter@colostate.edu
*    Correspondence: ryan.a.peterson@cuanschutz.edu

**Abstract:** Lockdowns implemented during the COVID-19 pandemic were utilized to evaluate the associations between "social distancing policies" (SDPs), traffic congestion, mobility, and NO$_2$ air pollution. Spatiotemporal linear mixed models were used on city-day data from 22 US cities to estimate the associations between SDPs, traffic congestion and mobility. Autoregressive integrated moving average models with Fourier terms were then used on historical data to forecast expected 2020 NO$_2$. Time series models were subsequently employed to measure how much reductions in local traffic congestion were associated with lower-than-forecasted 2020 NO$_2$. Finally, the equity of NO$_2$ pollution was assessed with community-level sociodemographics. When cities' most stringent SDPs were implemented, they observed a 23.47 (95% CI: 18.82–28.12) percent reduction in average daily congestion and a 13.48 (95% CI: 10.36–16.59) percent decrease in average daily mobility compared to unrestricted days. For each standard deviation (8.38%) reduction in local daily congestion, average daily NO$_2$ decreased by 1.37 (95% CI: 1.24–1.51) parts per billion relative to its forecasted value. Citizenship, education, and race were associated with elevated absolute NO$_2$ pollution levels but were not detectibly associated with reductions in 2020 NO$_2$ relative to its forecasted value. This illustrates the immediate behavioral and environmental impacts of local SDPs during the COVID-19 pandemic.

**Keywords:** COVID-19; air pollution; NO$_2$; traffic congestion; environmental equity; time series; forecasting

## 1. Introduction

Amidst the devastation of the COVID-19 pandemic, we harnessed the implementation of lockdowns and "social distancing policies" (SDPs) to study their impact on traffic congestion, mobility, nitrogen dioxide (NO$_2$) air pollution, and community-level demographics. Vehicle emissions are a primary source of exposure to NO$_2$, which is associated with adverse health and environmental impacts [1]. With traffic congestion concentrated in cities, near-road NO$_2$ is elevated [2] which puts urban residents at increased risk of developing asthma and reduced lung function and increases subsequent hospital admissions and mortality levels [3,4]. The implementation of SDPs related to the COVID-19 pandemic has decreased the number of workers who regularly commute. One survey estimates that over a third of workers transitioned from in-person to remote work in April 2020, while another 10 percent of workers were laid-off or furloughed [5]. Examining the extent to which SDPs impact mobility is important for evaluating the spread of COVID-19. This idea drove our interest in developing models to measure differential lockdown policy effects on mobility. In addition, given the ongoing climate crisis, we were motivated to track and estimate the effects of pandemic-related behavior change on air quality using novel statistical methods. Thus, our primary aims were to determine whether there is a

connection between city-level SDPs and local traffic congestion and identify the extent to which reductions in traffic congestion are associated with reductions in ambient $NO_2$ air pollution in US cities. Specifically, we sought to test the following primary hypotheses:

1. SDPs are associated with a reduction in average daily traffic congestion and mobility relative to pre-lockdown levels, after adjusting for snow, day of the week, federal holidays, seasonality, and autocorrelation within cities. Further, the reduction in congestion becomes more pronounced as SDPs become stricter. Note: We define 9 March 2020 as the start of the COVID-19 pandemic in the United States. "Pre-lockdown" refers to observations that occurred from 1 January 2019 to 8 March 2020. "Post-lockdown" refers to observations that occurred from 9 March 2020 to 31 August 2020.

2. There is a strong positive association between average daily traffic congestion and average daily $NO_2$ after adjusting for temperature, wind, and autocorrelation within cities.

3. Changes in average daily $NO_2$ levels observed post-lockdown are partially mediated by changes in daily average traffic congestion.

We also aimed to address the following secondary hypothesis concerning environmental equity:

4. Community-level sociodemographic factors including race, educational attainment, citizenship, and population density are associated with $NO_2$ air pollution. Furthermore, community demographics are associated with differences in the reduction in $NO_2$ air pollution during the COVID-19 pandemic relative to its forecasted value.

This work is organized as follows. First, we review gaps in recent and relevant literature that we aimed to address in this work. In Section 3, we describe our data sources, our decisions for categorizing lockdown policies across cities, and the statistical analysis methods we used to address our questions of interest. We then describe the results in Section 4, discuss their implications in Section 5, and summarize our conclusions in Section 6, and we end with our suggestions for future research in Section 7.

## 2. Literature Review

Existing works have investigated the local effects of pandemic-related changes in air pollution in singular locations. Zangari et al. (2020) found that in New York, lockdowns were associated with large decreases in $PM_{2.5}$ and $NO_2$, but these decreases became less pronounced or vanished when time series methods were used to account for temporal trends and variability [6]. However, their primary models measured the effect of the lockdown indirectly using a linear slope on time from 1 January of each year (2015–2020), which did not account for the timing or specificity of policies. In a California-specific study, Bashir et al. (2020) found a strong negative correlation between the number of cases and deaths due to COVID-19 and environmental pollution determinants; however, this work did not mention any correction for spatial or temporal dependence in their analysis [7]. Liu et al. (2021) did correct for temporal trends and accounted for lockdown timing in the Californian context, finding that sharp decreases in air pollutants occurred shortly after the lockdown policies were implemented [8]. This illustrates the important influence of lockdown timing. Another focused study in Seattle, WA, found through sophisticated time series analyses that local traffic decreased considerably while air quality improved during lockdown. They emphasized the importance of adjusting these analyses to take into account meteorological conditions [9]. On a broader scale, Chen et al. (2020) revealed that reductions in $NO_2$ and CO were common but heterogeneous across the US, after adjusting for temporal trends [10]. However, this analysis treated lockdown as a fixed period of time for each locale (starting and ending on the same date), and did not assess differential lockdown severity nor differences in mobility or traffic. Another US-based study by Berman & Ebisu (2020) found similar results on the county level; using two-sample aggregated comparisons, they found that urban counties saw especially large reductions in $NO_2$ after

13 March 2020 compared to historical data [11]. At the global level, Venter et al. (2020) connected mobility data with environmental indicators to argue that a strong link exists between global vehicle transportation declines and the reduction in ambient $NO_2$ exposure, though they did not assess how differences in lockdown policy measures differentially affected $NO_2$ [12].

Generally, these papers found that reductions in air pollution during the pandemic were most prominent for $NO_2$, which is notably influenced by car traffic [2]. While many other environmental indicators of air pollution (e.g., $PM_{2.5}$, $O_3$, CO) have been tied to adverse health outcomes (see Manisalidis et al. (2020) for a review [13]), a growing amount of evidence has specifically found adverse health effects from $NO_2$ pollution. $NO_2$ exposure has been associated with pediatric asthma [14], decreased lung capacity [15,16], and lung cancer [17]. The connection between $NO_2$ pollution and asthma is especially relevant during the COVID-19 pandemic. Evidence for asthma as a risk factor for severe COVID-19 has been surprisingly absent [18], while pediatric asthma health care utilization and prescribing has decreased throughout the pandemic [19].

Meanwhile, the pandemic has brought renewed efforts to design and implement technologies that have the potential to improve urban sustainability. These new and developing technologies include 6G, digital twins, and artificial intelligence, each of which can (or will soon) facilitate the use of large-scale data streams to simulate, measure, forecast, and streamline various city operations, ideally steering cities towards improved urban sustainability [20–22]. Unhealthy levels of air pollution caused by automobiles are incompatible with urban sustainability and therefore reducing automobile traffic is a concrete target for improving the health of our cities. Traffic congestion itself has been studied extensively, and several novel methods have recently been developed for describing and modeling such data, including dynamic route flow estimation, space-time autoregressive integrated moving average methods, and stochastic Markov models [23–25]. Finally, Benmarhnia (2020) discussed the implications of the pandemic on environmental equity, in particular citing a need for additional work in investigating the effects of different lockdown policies on historically disadvantaged communities [26].

Our study addresses several gaps in the current body of literature: we utilized sophisticated statistical methods to correct for spatial and temporal variability, accounted for local meteorological conditions, incorporated socioeconomic factors, sampled from a broad set of US cities, and formulated specific definitions to delineate the spectrum of lockdown policies which allowed us to measure the differential impact of specific lockdown policies. Further, our work is the first to our knowledge that explicitly measures the effects of these policies on environmental equity.

## 3. Research Methodology

### 3.1. Data Description

Daily data were collected from 1 January 2019 to 31 August 2020 from 22 major cities across all geographic regions of the United States. We assessed mobility with county-level Unacast electronic device GPS data [27] and congestion with TomTom metro-area traffic congestion data [28]. Local COVID-19 social distancing policies were obtained from the Washington Post (Johnson et al., 2020). $NO_2$ air quality data were collected by Core Based Statistical Area (CBSA) from the EPA [29,30] and weather data from each city's primary international airport were obtained from the NOAA [31]. Potential confounders, including wind and temperature, were identified based on previous literature [2,32,33]. Sociodemographic data were collected by ZIP Code Tabulation Area (ZCTA) [34].

### 3.2. Data Cleaning

#### 3.2.1. TomTom Traffic Congestion

TomTom traffic congestion data were reported as the percent change in average daily congestion from baseline. Baseline was defined as the average daily congestion by weekday and city, averaged over all days in 2019. This metric was used to model the

association between SDPs and congestion. TomTom data for 2020 were also reported as the average daily percent increase in the time that it took to drive with congestion compared to "free flow" travel without traffic. For interpretability, this metric was used to assess the association between congestion and average daily $NO_2$ levels. All TomTom data were missing on one day, so we imputed 22 (0.41%) daily observations by city for each TomTom metric using the Kalman smoothing method for time series models [35].

### 3.2.2. Unacast Mobility

Unacast mobility data were reported as the percent change in the average daily distance traveled from baseline. Baseline was defined as the average daily distance traveled by weekday and county during the four weeks prior to the start of the COVID-19 pandemic in the United States (10 February 2020 to 8 March 2020). The county that encompassed each city center was selected as a proxy for city-level mobility. (Note: Kansas City's mobility was averaged over the four counties that form its city center.)

### 3.2.3. Ambient $NO_2$

The latitude and longitude ranges of each city's CBSA [29,30] were identified to extract hourly $NO_2$ data by monitoring site from the EPA's AirNow API [36]. (Note: at the time of data collection, the validated data were not available. Thus, according to the EPA's documentation, these data should be considered preliminary as they have not yet been fully validated.) We first excluded four cities that did not have any $NO_2$ monitoring sites. Thirty-five additional monitoring sites that were missing more than 168 h (7 days) of contiguous data were eliminated. The eliminated sites had a median percent missingness of 28.75 (range: 2.98 to 88.14) percent. Missing hourly observations from the remaining 70 sites were imputed using the Kalman smoothing method for time series models [35]. We imputed 42,191 (4.07%) total hourly observations, with no more than 1597 (10.93%) imputed hours for any site. The hourly observations were then aggregated into mean daily concentrations by site. Ten $NO_2$ daily averages were negative due to zero drift [37], so we set these observations to zero. (Note: zero drift is a bias that occurs when the zero reading of a monitoring instrument is modified due to changing ambient conditions over time.) City-wide averages were then computed for 22 cities by averaging the mean daily concentration across monitoring sites (Figure 1).

### 3.2.4. NOAA Weather

The averages of daily high and low temperatures by city airport were used to create a daily temperature variable. Note: one daily low temperature was missing for one city (0.02% of daily low temperature observations across all cities), so it was imputed using the Kalman smoothing method [38]. Total daily snowfall was collapsed into "Heavy Snow" ($\geq$10 cm), "Light Snow" (<10 cm), and "No Snow". Due to data availability, wind was recorded as the fastest daily 2 min wind speed.

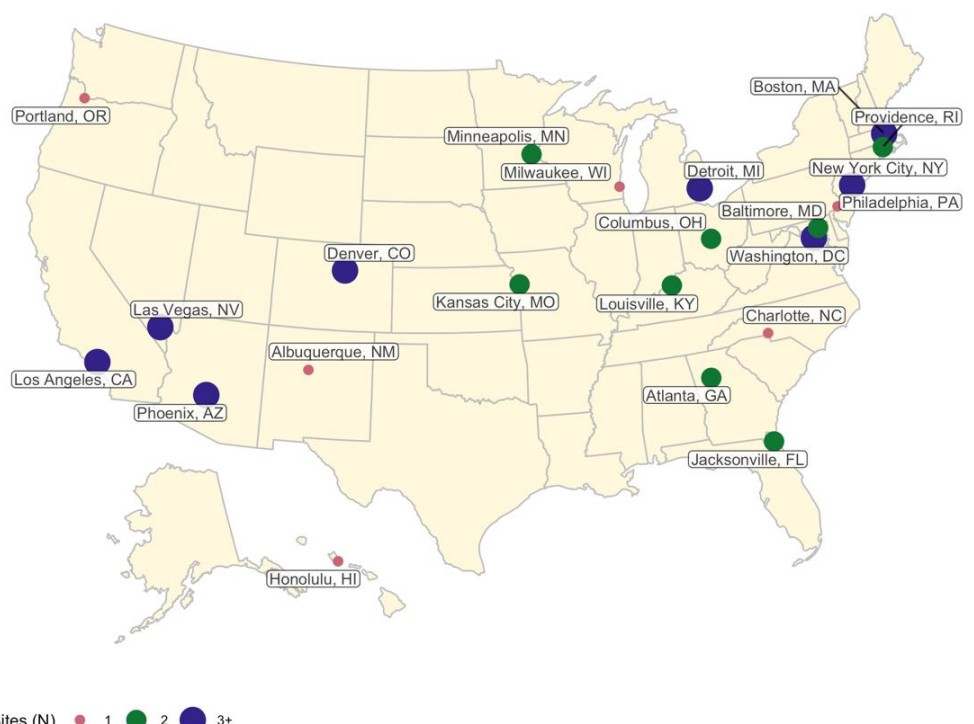

**Figure 1.** Available data. Cities used in the analysis are depicted on a map of the United States. The legend shows the number of NO$_2$ sites with sufficient data by city, where sufficient data were defined as having ≤168 consecutive hours (7 days) of missing observations. EPA NO$_2$, NOAA weather, and SDP data were collected from 1 January 2019 to 31 August 2020. TomTom traffic congestion data were available starting on 30 December 2019, and Unacast mobility data were available starting on 24 February 2020. Congestion and mobility data were collected through 31 August 2020.

### 3.2.5. COVID-19 Social Distancing Polices

Local COVID-19 social distancing policies (SDPs) were categorized into five restriction levels: "Closed", "Major Restrictions", "Moderate Restrictions", "Minor Restrictions", and "No Restrictions". These levels were based on policy news compiled by the Washington Post at the state level [39]. We modified the definitions to be more detailed and independently assigned restriction levels over time and by city (Figure 2). Additional news articles were used when the Washington Post indicated that policies varied within a state (Table S1).

### 3.2.6. Community Demographics

Information about educational attainment, race, citizenship, and population density were collected by ZCTA from the 2019 five-year Annual Community Survey (ACS) through the "tidycensus" API [40,41]. The educational attainment variables were "less than high school", "high school", "some college or an associate degree", and "at least a bachelor's degree" and race was categorized by whether people identified as "Asian", "Black", "White", "two or more races", or an "other race". Spatial data files were used to identify the latitude and longitude boundaries of each ZCTA. The coordinates of each NO$_2$ monitoring site were then used to determine its ZCTA [34]. Counts from overlapping ZCTAs were combined to estimate the demographic proportions of the communities near the monitoring sites. Counts were then converted into percentages of the total population of the surrounding community.

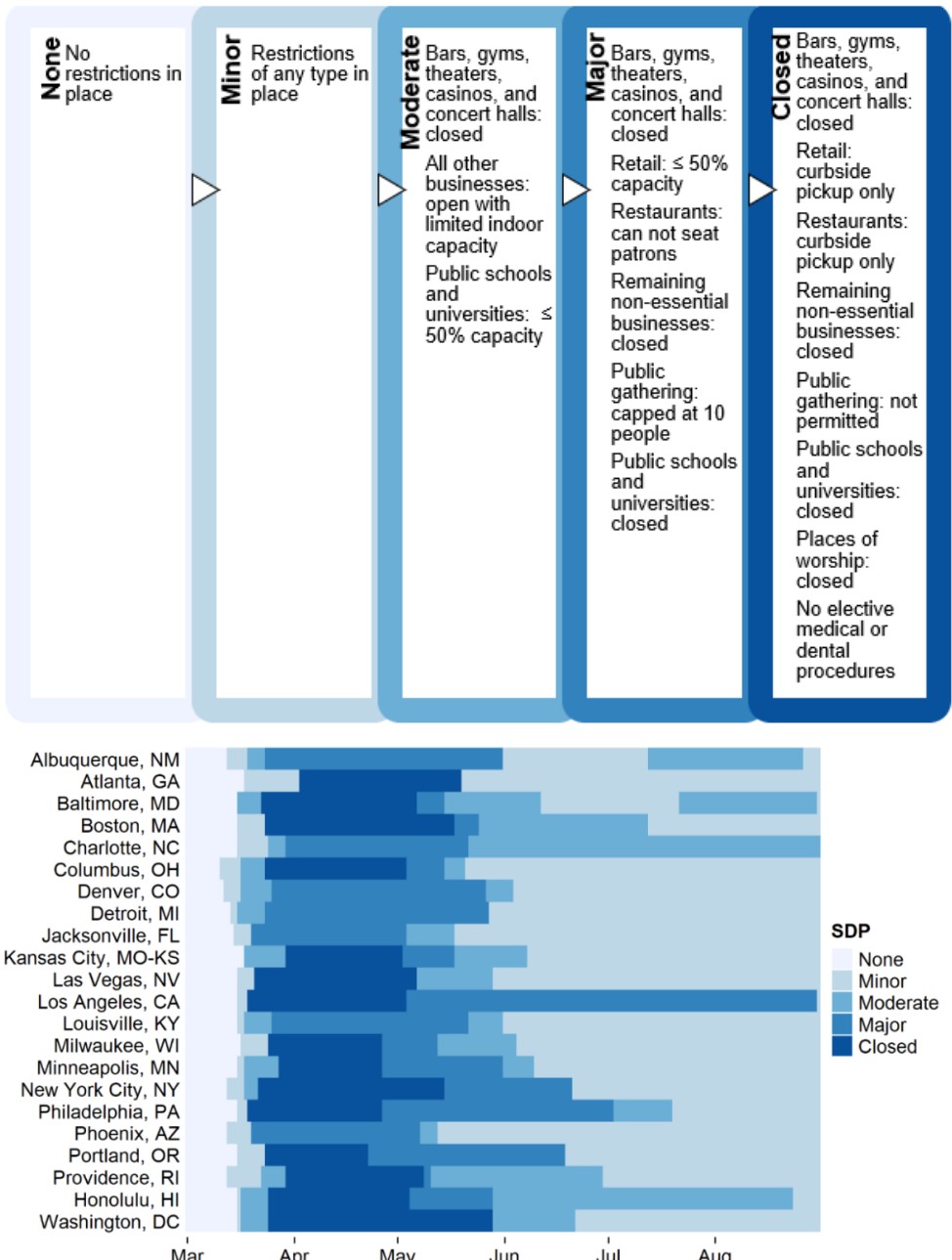

**Figure 2.** SDP definitions. If any criteria were violated, the city moved to the next less restrictive classification. For example, a city would be categorized as having "Major" SDPs if it met all "Closed" criteria except that places of worship were permitted to remain open. SDPs are presented from 1 March 2020 to 31 August 2020 by city. More stringent SDPs are presented in darker blue.

### 3.3. Regression Analyses

#### 3.3.1. Impact of SDPs on Congestion and Mobility

The association between SDPs and daily average congestion was first examined by performing a linear mixed model of the impact of SDPs on the percent change in TomTom average daily congestion from a 2019 baseline. We controlled for snow, day of the week and federal holidays. The SDP reference level was set to "No Restrictions" to assess the overall change in congestion following the implementation of COVID-19 lockdowns. An autoregressive moving average (ARMA(1,1)) structure was used to account for correlation within a city over time.

The same model structure was then used to assess the relationship between SDPs and the percent change in average daily distance traveled from the four weeks prior to COVID-19, as measured by Unacast mobility data.

Finally, to assess the relative differences in average daily congestion and mobility between minor and more stringent SDP restrictions, we reparametrized both models with minor restrictions as the SDP reference level.

### 3.3.2. Impact of Congestion on $NO_2$

The relationship between average daily congestion and $NO_2$ levels was assessed after controlling for wind and temperature. To address right-skewness, $NO_2$ was log transformed. We used a linear mixed model with a random intercept and an ARMA(1,1) covariance structure.

### 3.3.3. Impact of Congestion on Seasonally Adjusted $NO_2$

To determine whether the association between congestion and $NO_2$ held after accounting for seasonality, city-level average daily $NO_2$ from 1 January 2019 to 2 January 2020 was used to forecast expected average daily $NO_2$ from 3 January 2020 through 31 August 2020. Forecasting was completed using the "auto.arima" function from the R "forecast" package, which used stepwise selection to determine the best-fitting autoregressive integrated moving average (ARIMA) model according to the AICc [42]. This process utilized the Kwiatkowski–Phillips–Schmidt–Shin (KPSS) test to automatically assess and perform the minimum number of differences required for the series to be made stationary [43]. Two sine and cosine Fourier terms were included to account for seasonal trends. We then calculated the difference between the observed and forecasted $NO_2$ (seasonally adjusted $NO_2$). This was repeated for wind (seasonally adjusted wind) and temperature (seasonally adjusted temperature). Finally, we performed a linear mixed model of seasonally adjusted $NO_2$ regressed on congestion, seasonally adjusted wind, and seasonally adjusted temperature with a random intercept and ARMA(1,1) covariance structure.

### 3.3.4. Mediation Analysis

To estimate the impact of SDPs on seasonally adjusted $NO_2$, we performed a linear mixed model of seasonally adjusted $NO_2$ regressed on SDPs, seasonally adjusted wind, and seasonally adjusted temperature. We then added congestion to the model to assess how much the association between SDPs and seasonally adjusted $NO_2$ was mediated by congestion.

### 3.3.5. Measuring Equity in $NO_2$ Exposure across Community Demographics

A three-level hierarchical linear mixed model was used to assess the impact of education, race, citizenship, and population density on site-level $NO_2$, after adjusting for wind and temperature. To avoid multicollinearity, the percent of the population that had attained at least a bachelor's degree and the percent of the population that identified their race as "White" were not included in the model. The covariance structure included a random intercept for sites nested within cities and ARMA(1,1) repeated measures.

To determine the extent to which the change in $NO_2$ during COVID-19 was equitably distributed within a city, we repeated the $NO_2$ forecasting procedure with site-level $NO_2$ data. We then regressed seasonally adjusted $NO_2$ on all community-level sociodemographic variables, seasonally adjusted wind, and seasonally adjusted temperature. We used a three-level hierarchical linear mixed model with a random intercept for sites nested within cities and ARMA(1,1) repeated measures.

### 3.3.6. Model Selection

All regression models were implemented with a restricted maximum likelihood approach (REML). Model selection was not performed on the fixed effects, as confounders were included based on existing literature. AR(1) and ARMA(1,1) covariance structures

were considered to account for correlation within cities and a random intercept was considered to account for variability between cities. Covariance structures were selected based on BIC.

Data cleaning, descriptive statistics, and data visualization were performed in R version 4.0.2. Regression models were carried out in SAS 9.4.

## 4. Analysis and Results

### 4.1. Summary Statistics

Across all cities, 42.2 percent of post-lockdown city-day observations had minor restrictions in place, 17.0 percent had moderate restrictions in place, 20.6 percent had major restrictions in place, and 16.7 percent were closed (Table 1, Figure 2). Additionally, the median percent decrease in congestion from 2019 during the post-lockdown timeframe was 62.5 (IQR: 50.0 to 72.0) percent. The median percent decrease in mobility from the four weeks prior to COVID-19 was 28.8 (IQR: 17.2 to 43.1) percent. The median post-lockdown daily average $NO_2$ concentration was 8.18 (IQR: 5.49 to 11.20) ppb, and the median pre-lockdown average daily $NO_2$ concentration was 11.30 (IQR: 7.66 to 16.10) ppb (Table 1).

**Table 1.** Summary statistics averaged over 22 US cities.

| | Pre-Lockdown [1] (N = 9526) [3] | Post-Lockdown [2] (N = 3872) |
|---|---|---|
| **SDPs** | | |
| No Restrictions | 9526 (100.0%) | 135 (3.5%) |
| Minor Restrictions | 0 (0.0%) | 1634 (42.2%) |
| Moderate Restrictions | 0 (0.0%) | 658 (17.0%) |
| Major Restrictions | 0 (0.0%) | 798 (20.6%) |
| Closed | 0 (0.0%) | 647 (16.7%) |
| **TomTom: % Change in Congestion from Baseline** | | |
| N | 1540 | 3872 |
| Mean (SD) | −3.9 (25.9) | −60.1 (17.6) |
| Median (Q1, Q3) | −4.2 (−16.7, 8.3) | −62.5 (−72.0, −50.0) |
| **Unacast: % Change in Mobility from Baseline** | | |
| N | 308 | 3872 |
| Mean (SD) | 0.8 (4.2) | −30.5 (17.1) |
| Median (Q1, Q3) | 0.8 (−2.0, 3.4) | −28.8 (−43.1, −17.2) |
| **Daily Average $NO_2$ Concentration (ppb)** | | |
| N | 9526 | 3872 |
| Mean (SD) | 12.5 (6.6) | 8.6 (4.1) |
| Median (Q1, Q3) | 11.3 (7.7, 16.1) | 8.2 (5.5, 11.2) |
| **Snow [4]** | | |
| Heavy Snow | 39 (0.4%) | 2 (0.1%) |
| Light Snow | 393 (4.1%) | 27 (0.7%) |
| No Snow | 9094 (95.5%) | 3843 (99.3%) |
| **Daily Temperature (°C) [5]** | | |
| N | 9526 | 3872 |
| Mean (SD) | 13.8 (10.6) | 20.4 (8.0) |
| Median (Q1, Q3) | 14.2 (5.5, 22.8) | 22.2 (14.7, 26.4) |
| **Fastest Daily 2 Minute Wind Speed (m/s)** | | |
| N | 9526 | 3872 |
| Mean (SD) | 8.6 (2.9) | 8.9 (2.8) |
| Median (Q1, Q3) | 8.1 (6.3, 10.3) | 8.1 (6.7, 10.3) |

[1] "Pre-Lockdown" refers to any observation from 1 January 2019 to 8 March 2020. The earliest TomTom congestion observation was on 30 December 2019 and the earliest Unacast mobility observation was on 24 February 2020. The first observations for all other variables were on 1 January 2019. [2] "Post-Lockdown" refers to any observation that occurred between 9 March 2020 and 31 August 2020. [3] Observations (N) are reported at the city-day level, such that one observation represents data from one day within one city. Since there are 22 cities in the dataset, each day of data has 22 observations. [4] "Heavy Snow" is defined as any total daily snowfall greater than or equal to 10 cm, and "Light Snow" is defined as any total daily snowfall of less than 10 cm. [5] Daily temperature is calculated as the mean of the daily minimum temperature and the daily maximum temperature.

### 4.2. Impact of SDPs on Congestion and Mobility

When minor SDPs were implemented, cities observed a 17.96 (95% CI: 14.56 to 21.36) percent reduction in average daily congestion compared to unrestricted days, after adjusting for wind, temperature, the variability between cities, and the correlation within cities over time ($p < 0.001$). Days that had moderate restrictions in place had a 20.94 (95% CI: 17.01 to 24.87) percent reduction in average daily congestion and days that had major restrictions in place had a 20.66 (95% CI: 16.14 to 25.17) percent reduction in average daily congestion compared to unrestricted days ($p < 0.001$; $p < 0.001$). Days that had closed restrictions in place experienced the largest decrease in congestion; they observed a 23.47 (95% 18.82 to 28.12) percent reduction in average daily congestion compared to unrestricted days ($p < 0.001$). As expected, congestion was negatively associated with federal holidays. Heavy snow was associated with an increase in congestion, while light snow did not significantly impact congestion. Weekdays were also a strong predictor of mobility, despite their inclusion in the baseline reference, which was defined as the average daily congestion by weekday and city, averaged over all days in 2019 (Table 2).

**Table 2.** TomTom congestion and Unacast mobility regressed on SDPs.

| Effect | TomTom Congestion | | Unacast Mobility | |
|---|---|---|---|---|
| | Estimate (95% CI) | *p*-Value | Estimate (95% CI) | *p*-Value |
| Intercept | −21.95 (−28.23, −15.66) | <0.001 | −17.92 (−22.04, −13.8) | <0.001 |
| SDP: Minor | −17.96 (−21.36, −14.56) | <0.001 | −5.65 (−7.9, −3.39) | <0.001 |
| SDP: Moderate | −20.94 (−24.87, −17.01) | <0.001 | −8.17 (−10.81, −5.54) | <0.001 |
| SDP: Major | −20.66 (−25.17, −16.14) | <0.001 | −9.68 (−12.7, −6.67) | <0.001 |
| SDP: Closed | −23.47 (−28.12, −18.82) | <0.001 | −13.48 (−16.59, −10.36) | <0.001 |
| SDP: None | Reference | | | |
| Heavy Snow | 38.88 (29.46, 48.3) | <0.001 | −12.34 (−17.94, −6.73) | <0.001 |
| Light Snow | −0.54 (−3.45, 2.37) | 0.715 | −2.2 (−3.96, −0.44) | 0.014 |
| Sunday | −4.14 (−5, −3.29) | <0.001 | −2.68 (−3.18, −2.17) | <0.001 |
| Monday | −8.62 (−9.53, −7.72) | <0.001 | −0.62 (−1.17, −0.07) | 0.028 |
| Tuesday | −11.96 (−12.89, −11.04) | <0.001 | 1.74 (1.17, 2.31) | <0.001 |
| Wednesday | −11.46 (−12.39, −10.53) | <0.001 | 1.37 (0.8, 1.94) | <0.001 |
| Thursday | −10.52 (−11.43, −9.62) | <0.001 | 0.27 (−0.28, 0.82) | 0.336 |
| Friday | −6.49 (−7.36, −5.63) | <0.001 | −1.3 (−1.8, −0.79) | <0.001 |
| Saturday | Reference | | | |
| Federal Holiday | −13.27 (−15.76, −10.79) | <0.001 | −1.52 (−3, −0.04) | 0.044 |

Findings from the linear mixed model regression analyses of the impact of SDPs on TomTom congestion and Unacast mobility are presented. TomTom congestion estimates are interpreted in terms of the percent change in average daily traffic congestion compared to unrestricted days, where positive estimates indicate an increase in congestion and negative estimates indicate a reduction in congestion. Unacast mobility estimates are interpreted in terms of the percent change in the average daily distance traveled compared to unrestricted days, where positive estimates indicate an increase in mobility and negative estimates indicate a reduction in mobility.

There was significantly less congestion when more stringent SDPs were implemented compared to days when minor SDPs were implemented. Days that had moderate restrictions in place saw a 2.98 (95% CI: 0.44 to 5.51) percent decrease in average daily congestion compared to days when minor restrictions were implemented ($p = 0.022$). When closed restrictions were implemented, cities observed a 5.51 (95% 2.03 to 8.99) percent reduction in average daily congestion compared to minor restrictions ($p = 0.002$). Major restrictions were not significantly different from minor restrictions ($p = 0.108$) (Table S2).

As expected, SDPs were also associated with reductions in mobility. When minor SDPs were implemented, cities observed a 5.65 (95% CI: 3.39 to 7.90) percent reduction in average daily mobility compared to unrestricted days, after adjusting for wind, temperature, the variability between cities, and the correlation within cities over time ($p < 0.001$). Cities observed an 8.17 (95% CI: 5.54 to 10.81) percent decline in average daily mobility when moderate restrictions were implemented and a 9.68 (95% CI: 6.67 to 12.70) percent decline in average daily mobility when major restrictions were implemented compared to unrestricted days ($p < 0.001$; $p < 0.001$). Days that had closed restrictions in place observed a 13.48 (95% CI 10.36 to 16.59) percent reduction in average daily mobility compared to unrestricted days ($p < 0.001$). Federal holidays and heavy snow were associated with decreases in mobility,

while light snow was a non-significant predictor of mobility. Weekdays were often a significant predictor of mobility despite their inclusion in the baseline reference, which was defined as the average daily distance traveled by weekday and county during the four weeks prior to the start of the COVID-19 pandemic in the United States (10 February 2020 to 8 March 2020) (Table 2).

More stringent SDPs were also associated with less mobility relative to when minor restrictions were implemented. Days that had moderate restrictions in place saw a 2.53 (95% CI: 0.85 to 4.2) percent decrease in the average daily distance traveled relative to unrestricted days ($p = 0.004$). Cities observed a 4.04 (95% CI: 1.96 to 6.22) percent reduction in average daily mobility under major restrictions and a 7.83 (95% CI: 5.52 to 10.14) percent reduction in average daily mobility under closed restrictions ($p < 0.001$; $p < 0.001$) (Table S2).

### 4.3. Impact of Congestion on $NO_2$

Average daily $NO_2$ decreased by 22.71 (95% CI: 21.15 to 24.29) percent for every standard deviation (8.36 percent) decrease in the average daily time it took to drive with traffic compared to free flow ($p < 0.001$). Additionally, for every standard deviation (2.91 m per second) increase in the fastest daily 2 min wind speed, average daily $NO_2$ decreased by 10.29 (95% CI: 9.48 to 11.08) percent ($p < 0.001$). A standard deviation (10.10 °C) increase in average daily temperature was associated with a 4.61 (95% CI: 3.16 to 6.03) percent decrease in average daily $NO_2$ ($p < 0.001$) (Table 3).

**Table 3.** $NO_2$ regressed on TomTom congestion.

| Effect | Estimate (95% CI) | *p*-Value |
|---|---|---|
| Intercept | 11.2633 (9.7794, 12.9723) | <0.001 |
| TomTom Congestion | 1.0248 (1.0232, 1.0263) | <0.001 |
| Wind (m/s) | 0.9634 (0.9605, 0.9663) | <0.001 |
| Temperature (°C) | 0.9953 (0.9939, 0.9968) | <0.001 |

Findings from the linear mixed model regression analysis of the log of $NO_2$ regressed on TomTom congestion are reported. TomTom congestion is reported as the average daily percent increase in the time that it took to drive with congestion compared to "free flow" travel without traffic. Seasonally adjusted wind is reported in meters per second, and seasonally adjusted temperature is reported in degrees Celsius. Results are interpreted in terms of percent changes in $NO_2$. Exponentiated estimates greater than one indicate an increase in $NO_2$ associated with a one unit increase in the predictor variable and exponentiated estimates less than one indicate a decrease in $NO_2$ associated with a one unit increase in the predictor variable.

### 4.4. Impact of Congestion on Seasonally Adjusted $NO_2$

After accounting for seasonality, congestion remained strongly associated with $NO_2$. For every standard deviation (8.38 percent) decrease in the average daily time it took to drive with traffic compared to free flow during 2020, average daily $NO_2$ was 1.37 (95% CI: 1.24 to 1.51) ppb lower than forecasted, after accounting for seasonally adjusted wind and seasonally adjusted temperature ($p < 0.001$). Additionally, for each standard deviation (2.72 m per second) increase in the difference between the observed and forecasted fastest daily two-minute wind speed, average daily $NO_2$ was 1.22 (95% CI: 1.12 to 1.31) ppb lower than forecasted ($p < 0.001$). Each standard deviation (3.76 °C) increase in the difference between the observed and forecasted temperature was associated with a 0.38 (95% CI: 0.26 to 0.50) ppb increase in $NO_2$ relative to forecasted ($p < 0.001$) (Table 4). (Note: past research has shown that temperature has a negative association with $NO_2$ in the winter and a positive association with $NO_2$ in the summer [43]. Since our data included more observations that occurred in the summer than in the winter, this result was expected.) The association between congestion, mobility, and $NO_2$ is further illustrated in Figure 3 and Figure S1.

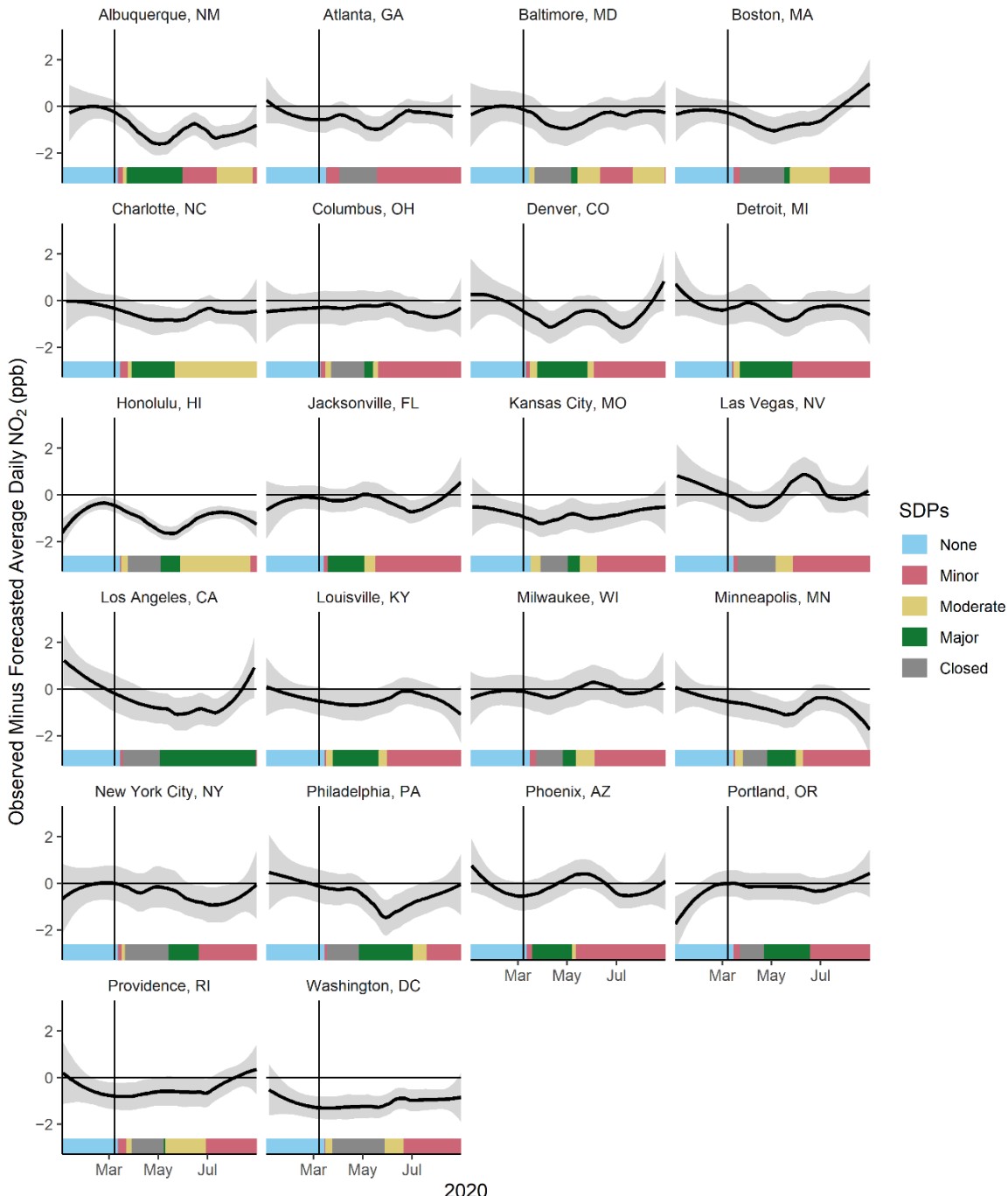

**Figure 3.** Seasonally adjusted average daily $NO_2$ (ppb) by city over time. The difference in ppb between observed and ARIMA-forecasted 2020 ambient $NO_2$ levels is plotted by city using LOESS smoothing with a span of 0.6. The 95% confidence interval around the LOESS curve is in gray and the rug is color-coded based on city-specific SDPs over time. Vertical lines signify the start of the COVID-19 pandemic in the United States on 9 March 2020. Observations above the horizontal line indicate that observed $NO_2$ was greater than forecasted and observations below the horizontal line indicate that observed $NO_2$ was lower than forecasted. See Figure S1 for more detailed data and Figure S2 for similar city-level changes in traffic congestion.

**Table 4.** Seasonally adjusted $NO_2$ regressed on congestion.

| Effect | Estimate (95% CI) | *p*-Value |
|---|---|---|
| Intercept | −3.2143 (−3.5741, −2.8544) | <0.001 |
| TomTom Congestion | 0.1637 (0.1478, 0.1797) | <0.001 |
| Seasonally Adjusted Wind (m/s) | −0.4485 (−0.4831, −0.4139) | <0.001 |
| Seasonally Adjusted Temp (°C) | 0.1000 (0.0679, 0.1321) | <0.001 |

The above table presents findings from the linear mixed model regression analysis of the impact of TomTom congestion on seasonally adjusted average daily $NO_2$. TomTom Congestion is reported as the average daily percent increase in the time that it took to drive with congestion compared to "free flow" travel without traffic. Seasonally adjusted wind is the difference between the observed and forecasted wind speed in meters per second, and seasonally adjusted temperature is reported as the difference between observed and forecasted temperature in degrees Celsius. Estimates are interpreted in terms of ppb changes in $NO_2$ relative to forecasted.

*4.5. Mediation Analysis*

There was a strong association between more stringent SDPs and lower levels of seasonally adjusted $NO_2$, after adjusting for seasonally adjusted wind and seasonally adjusted temperature. We then added congestion to the model to examine its mediating impact; the effect of SDPs on $NO_2$ was completely mediated by changes in congestion, such that more stringent SDPs were associated with a greater increase in $NO_2$ relative to its forecasted value. According to the mediation model, cities observed a 2.13 (95% CI: 1.70 to 2.55) ppb increase in average daily $NO_2$ when minor restrictions were implemented compared to unrestricted days, after adjusting for congestion, seasonally adjusted wind, and seasonally adjusted temperature ($p < 0.001$). Similarly, cities observed a 1.52 (95% CI: 0.98 to 2.06) ppb increase in average daily $NO_2$ when moderate restrictions were implemented and a 1.29 (95% CI: 0.76 to 1.82) ppb increase in average daily $NO_2$ when major restrictions were implemented compared to unrestricted days ($p < 0.001$; $p < 0.001$). Days that had closed restrictions in place observed a 1.45 (95% CI: 0.88 to 2.03) ppb increase in average daily $NO_2$ relative to unrestricted days, after adjusting for congestion ($p < 0.001$).

*4.6. Measuring Equity in $NO_2$ Exposure across Community Demographics*

Sociodemographic patterning of $NO_2$ pollution was evident; each standard deviation (7.22 percent) increase in non-citizens was associated with a 29.97 (95% CI: 11.42 to 51.62) percent increase in $NO_2$, after adjusting for wind, temperature, education, race, and population density ($p = 0.001$). No other social or demographic factors were significant predictors of $NO_2$ in the full multivariable model (Table 5). However, when each sociodemographic variable was considered individually, we found that each standard deviation (12.12 percent) increase in community members with less than a high school education was associated with a 16.97 (95% CI: 4.86 to 30.48) percent increase in $NO_2$ ($p = 0.022$). Additionally, each standard deviation (3.48 percent) increase in individuals who identified with two or more races was associated with a 11.81 (95% CI: 2.98 to 19.84) percent decrease in $NO_2$ ($p = 0.010$) and each standard deviation (11.76 percent) increase in people who identified as "other race" was associated with a 18.89 (95% CI: 8.32 to 30.48) percent increase in $NO_2$ ($p < 0.001$).

No demographic variables were significantly associated with seasonally adjusted $NO_2$, according to the full multivariable model (Table S3). When each demographic variable was added to the model individually, each standard deviation (7.91 percent) increase in community members with a high school education was associated with a 0.04 (95% CI: 0.00 to 0.08) ppb greater reduction in $NO_2$ than forecasted ($p = 0.045$) and each standard deviation (6.10 percent) increase in the population with some college education was associated with 0.07 (95% CI: 0.02 to 0.11) ppb less of a reduction in $NO_2$ relative to forecasted ($p = 0.003$). A higher population density was also associated with a greater reduction in $NO_2$ than forecasted ($p = 0.008$); however, the effect size was too small to be practically meaningful; a standard deviation increase in the population density (2.29 thousand people per square kilometer) was associated with a 0.0003 (95% CI: 0.0001 to 0.0005) ppb greater reduction in $NO_2$ than forecasted ($p = 0.008$) (Table S3).

**Table 5.** $NO_2$ regressed on community-level demographics.

| Effect | Estimate (95% CI) | *p*-Value |
|---|---|---|
| Intercept | 8.2644 (5.0422, 13.5458) | <0.001 |
| Wind | 0.9627 (0.9607, 0.9648) | <0.001 |
| Temperature | 0.993 (0.9917, 0.9942) | <0.001 |
| Race: Black | 1.0018 (0.9972, 1.0065) | 0.442 |
| Race: Asian | 0.9966 (0.9829, 1.0104) | 0.627 |
| Race: Two or More | 0.9813 (0.955, 1.0084) | 0.174 |
| Race: Other | 1.0013 (0.9894, 1.0133) | 0.836 |
| Education: Less Than High School | 0.9961 (0.9802, 1.0123) | 0.636 |
| Education: High School | 1.0112 (0.9951, 1.0275) | 0.173 |
| Education: Some College | 1.0027 (0.9847, 1.0211) | 0.767 |
| Non-Citizen | 1.037 (1.0151, 1.0594) | 0.001 |
| Population Density | 1.0091 (0.9659, 1.0543) | 0.684 |

The above table presents findings from the linear mixed model regression analysis of the log of $NO_2$ regressed on wind, temperature, education, race, citizenship, and population density. TomTom congestion is reported as the average daily percent increase in the time that it took to drive with congestion compared to "free flow" travel. The population density is in terms of the number of thousands of people per square kilometer. Results are interpreted in terms of percent changes in $NO_2$. Exponentiated estimates greater than one indicate an increase in $NO_2$ associated with a one unit increase in the predictor variable and exponentiated estimates less than one indicate a decrease in $NO_2$ associated with a one unit increase in the predictor variable.

## 5. Discussion

Our analysis identified and meticulously measured important associations between SDPs, congestion, mobility, and $NO_2$ pollution. We found a strong association between more stringent SDPs and reductions in both traffic congestion and mobility. There was also a strong positive relationship between reductions in congestion and reductions in $NO_2$ relative to its forecasted value. Furthermore, the effect of SDPs on reductions in $NO_2$ was largely mediated by congestion.

There was substantial heterogeneity across cities in terms of congestion, mobility, and $NO_2$. Cities followed markedly different trends in pre-lockdown reductions in $NO_2$ relative to its forecasted value, although this may have been driven by greater pre-lockdown variability in $NO_2$ (Figure 3, Figures S1 and S2). Shortly following the start of lockdowns, most cities then experienced a steep drop in congestion, mobility, and $NO_2$. Overall, these measures slowly increased toward pre-lockdown levels through August 2020, but the effect size and timing of these trends varied across cities. This heterogeneity contributed to considerably smaller within-city fixed effect estimates in the linear mixed model of the impact of SDPs on congestion compared to unadjusted estimates (Tables 1 and 2).

We also gained insight into the sociodemographic patterning of $NO_2$ pollution. One notable finding was a strong positive association between communities with a higher proportion of non-citizens and absolute $NO_2$ levels, after adjusting for other sociodemographic factors, wind, and temperature. While variation in $NO_2$ pollution was best captured by citizenship, communities with lower overall educational attainment opportunities and higher proportions of people who identified as an "other race" were also strongly associated with elevated absolute $NO_2$. Future research must disaggregate the "other race" category to provide a more nuanced understanding of which communities experience elevated $NO_2$ pollution. Our data show that disadvantaged communities were indeed exposed to greater levels of $NO_2$ than their counterparts. With known negative health impacts associated with $NO_2$ exposure, this is a fundamental violation of environmental justice [26].

Encouragingly, we did not find evidence to conclude that these existing inequities in exposure to $NO_2$ pollution were exacerbated post-lockdown. According to the full multivariate model of the impact of demographic variables on seasonally adjusted $NO_2$, the extent to which $NO_2$ was lower than forecasted was not significantly associated with any of the demographic variables we were able to include. When each sociodemographic factor was considered individually, communities disadvantaged by lower educational

attainment had a greater reduction in $NO_2$ than communities with higher educational attainment. However, the effect size of this association was negligible.

Our results must be considered within the context of important limitations. Firstly, each data source used different geographic bounds. For example, congestion was reported from an unspecified metropolitan boundary, while $NO_2$ monitoring sites were located in both urban centers and the surrounding suburbs. Since there may be a greater association between $NO_2$ and congestion on the roads closest to the monitoring sites, using metro area data may have reduced the association of our findings. Likewise, community-level sociodemographic data were collected from variably sized boundaries. There may have been greater associations between sociodemographic variables and $NO_2$ if the sizes of the boundaries surrounding the monitoring sites were reduced. Our analysis was also limited by the potential for unmeasured confounders, such as construction or widespread protests during the summer of 2020. We were also limited by the consistency of available EPA monitoring sites. Some cities were missing $NO_2$ monitoring sites entirely, and we considered other sites to be non-representative because they were missing more than a week of consecutive hourly observations. However, even with missingness, we created a comprehensive dataset with 70 sites across 22 US cities.

Additionally, we designed an ordinal scale to categorize local social distancing policies by their restrictiveness. There was a level of subjectivity in these definitions because risk assessments for various activities within the context of the COVID-19 pandemic have yet to be thoroughly established. However, by developing a consistent scale across cities, we minimized bias in the SDP definitions. While previous research has examined the impact of lockdowns on congestion and $NO_2$, our SDP scale allowed us to granularly examine the impact of policy changes over time and across cities. By using detailed time series models, we had the sensitivity to evaluate more nuanced changes in SDPs, and their subsequent associations with congestion, mobility, and $NO_2$.

## 6. Conclusions and Recommendations

We found that lockdown policies in cities across the US led to substantial, heterogeneous reductions in traffic and mobility, which were followed by considerable reductions in air pollution. The observed heterogeneity in these markers was likely reflective of policy decisions made at the local level. When minor SDPs (e.g., restrictions less severe than closing non-essential businesses) were implemented, cities observed nearly a 20 percent reduction in average daily congestion and a 6 percent reduction in average daily mobility compared to unrestricted days. Days that had the most stringent restrictions in place (e.g., closing non-essential businesses, halting elective medical procedures, and shutting down public schools, places of worship, and gatherings) observed a 24 percent reduction in average daily congestion and nearly a 14 percent reduction in average daily mobility compared to unrestricted days. Subsequently, declines in the average daily time it took to drive with traffic corresponded to lower average daily $NO_2$ in 2020 relative to forecasted (Table 4). Furthermore, the effect of SDPs on reductions in $NO_2$ was completely mediated by changes in congestion. This finding suggests that even minor policy changes might meaningfully reduce traffic congestion and $NO_2$ pollution.

Additionally, we found evidence that sociodemographic inequities in $NO_2$ exposure have persisted (but were not necessarily worsened) during the COVID-19 pandemic; for example, a moderate (one standard deviation) increase in the local proportion of non-citizens was associated with almost a 30% increase in $NO_2$. This result highlights the importance of targeting future policies towards places and communities with higher exposure to $NO_2$.

## 7. Future Research

As more of the population begins to commute again, future research should assess the lasting impact of congestion on $NO_2$. With heterogeneity in the timing and restrictiveness of SDPs across cities, our research demonstrates that local policies are associated

with modifications in the public's behavior. This finding justifies policies targeting more sustainable urban planning, such as infrastructure and incentives for green commuting. Furthermore, our results suggest that SDPs may have a diminishing marginal return on congestion; minor restrictions were associated with a large decrease in congestion, but as policies became stricter, the additional reduction in congestion decreased. If this relationship can be confirmed in future research, it would follow that even minor policy changes might meaningfully reduce traffic congestion and $NO_2$ pollution. Furthermore, future research should collect and analyze more granular sociodemographic variables by disassembling the heterogeneous "other" category such that less-represented groups are included in policies targeting equitable protection from environmental hazards. Finally, while we have shown important relationships between SDPs, traffic congestion, and $NO_2$, it would be pertinent to use similar analysis methods to investigate the extent to which other air pollutants (e.g., $PM_{2.5}$, $CO$, $O_3$) are related to SDPs in future broad-scale studies.

**Supplementary Materials:** The following are available online at https://www.mdpi.com/article/10.3390/su13137275/s1, Table S1. SDP additional news sources, Table S2. Alternative parameterization of congestion and mobility regressed on SDPs, Table S3. Seasonally adjusted NO2 and sociodemographics, Figure S1. Observed minus forecasted average daily NO2 (ppb) by city over time, Figure S2. City-level changes in traffic congestion.

**Author Contributions:** Conceptualization, A.K.W., R.A.P. and M.D.S.; methodology, A.K.W., R.A.P. and M.D.S.; software, A.K.W. and R.A.P.; validation, A.K.W., R.A.P. and M.D.S.; formal analysis, A.K.W.; data curation, A.K.W.; writing—original draft preparation, A.K.W.; writing—review and editing, R.A.P., E.C. and M.D.S.; visualization, A.K.W.; supervision, R.A.P. and M.D.S.; project administration, R.A.P. All authors have read and agreed to the published version of the manuscript.

**Funding:** This research received no external funding.

**Institutional Review Board Statement:** Not applicable.

**Informed Consent Statement:** Not applicable.

**Data Availability Statement:** Unacast (2020) data were obtained from Unacast and are available from the authors at https://www.unacast.com/data-for-good with the permission of Unacast. Data from Menne et al. (2012) are openly available at doi:10.7289/V5D21VHZ. The following data are publicly available and can be found here: www.tomtom.com/en_gb/traffic-index; https://aqs.epa.gov/aqsweb/documents/codetables/cbsas.html; https://aqs.epa.gov/aqsweb/airdata/download_files.html; https://docs.airnowapi.org/Data/docs; https://www.census.gov/geographies/mapping-files/2018/geo/carto-boundary-file.html.

**Conflicts of Interest:** The authors declare no conflict of interest.

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
