# Peer review of "Impact of COVID-19 Social Distancing Policies on Traffic Congestion, Mobility, and NO2 Pollution"

_sustainability, doi:10.3390/su13137275_

Round 1
Reviewer 1 Report
The paper addresses an interesting concept," Impact of COVID-19 Social Distancing Policies on Traffic Congestion, Mobility, and NO2 Pol". The authors have invested considerable thought and effort into the problem investigated. The work is almost well-understood and falls within the scopes defined in the journal, though in some parts, the writing deteriorates considerably and needs some amendments. Overall, the paper is interesting from the practical and theoretical points of view. There are, however, some deficiencies with the paper, which should be addressed/responded to before it can be recommended for publication in the Journal of Sustainability.
* Introduction section needs to be more comprehensive. The authors need to review more related articles in this section. In addition, the authors need to describe shortly how they created these Hypotheses. Also, I suggest the authors add a paragraph at the end of this section to show the paper structure at the end of the Introduction.
* I strongly suggest having a Background or Literature review section after the Introduction section. In this section, the authors need to review all main related keywords.
* In the Introduction or Background sections, the authors need to review some technologies which can help model, simulate, analyze, and predict Traffic Congestion, Mobility, and NO2 Pollution (such as Building Information Modeling, Digital Twins, and AI). Hoping to help in this task, I provide some literature:
- Future (post-COVID) digital, smart and sustainable cities in the wake of 6G: Digital twins, immersive realities and new urban economies
- Towards a Digital Twin-based Smart Built Environment
- Revisiting the built environment: 10 potential development changes and paradigm shifts due to COVID-19
- Digital twins based on bidirectional LSTM and GAN for modelling the COVID-19 pandemic
- Digital twin-based progress monitoring management model through reality capture to extended reality technologies (DRX)
- National energy and climate planning approach for the Western Balkans: case study Republic of Serbia
- Dynamic Route Flow Estimation in Road Networks Using Data from Automatic Number of Plate Recognition Sensors
* The quality of figure 1 needs to be improved.
* You need to add the Conclusion section after the Discussion section.
* Please add future research after the Conclusion section.
Reviewer 2 Report
Thank you for the manuscript. The authors used data from 22 cities in the United States to study the impact of COVID-19 Social Distancing Policies on Traffic Congestion, Mobility, and NO2 Pollution which is interesting. Comments to authors:
1)The paper uses time series analysis but lack of robustness test. I suggest that the author add stationarity and cointegration tests to avoid spurious regression.
2)lockdown is obvious to reduce the use of cars, it would be better if the differences in the implementation of lockdown policies in different cities can be summarized.
Round 2
Reviewer 1 Report
I would like to thank the authors for the great improvements they made to the original manuscript. However, some minor revisions need to be done before submitting the paper to the journal.
- Your section numbers are not correct. 1. Introduction, 2. Literature review, 3. Research methodology, 4. Analysis and results, 5. Discussion, 6. Conclusion and recommendations, 7. Future Research
- The Conclusion section needs to be more comprehensive—at least two or three paragraphs.
Round 3
Reviewer 1 Report
Accept in present form.